# OpenReview forum: "SciKnowEval: Evaluating Multi-level Scientific Knowledge of Large Language Models"
_ICLR.cc/2025/Conference — Submitted to ICLR 2025_

### Official Review · Reviewer_R2P1 · 2024-11-03

**Soundness:** 3
**Presentation:** 3
**Contribution:** 2
**Rating:** 6
**Confidence:** 4

**Summary:**

This paper presents SciKnowEval, a novel benchmark framework designed to assess the scientific knowledge capabilities of large language models (LLMs). Its distinctive five-level knowledge philosophy encompasses a broad spectrum of knowledge, ranging from knowledge memory to knowledge utilization. The dataset is meticulously curated from diverse sources and employs a combination of LLM screening techniques and sampled human evaluation to ensure data quality and efficiency. Through the evaluation of both proprietary and open-source models using SciKnowEval, the paper demonstrates that sota models still have considerable room for improvement in their scientific knowledge capabilities.

**Strengths:**

The dataset curated by the authors encompasses various facets of scientific knowledge, allowing for the evaluation of models in specific aspects of science. This enables a more granular understanding of how models perform at different levels, from knowledge awareness to knowledge application. The dataset's size is relatively significant within its domain, and the quality control measures employed, involving both human involvement for sampled questions and LLM screening for all questions, enhance efficiency while maintaining credibility.

**Weaknesses:**

While the benchmark aims to assess scientific reasoning abilities, it's crucial to acknowledge that QA-like tasks might not be comprehensive enough to evaluate models' scientific reasoning capabilities, even with categorized levels. Assessing models' scientific reasoning capabilities may require more complex setups, including tool usage, task planning, and end-to-end experiment conducting. It's worth mentioning these limitations in order to provide a more nuanced expectation for readers. For example, the authors can add a paragraph or short description as a limitation section, acknowledging that while their benchmark provides valuable insights, fully assessing scientific reasoning may require more complex setups. This would help set more accurate expectations for readers about what the benchmark can and cannot evaluate.

Nitpicks:
Make sure left quotation marks are used correctly, such as in table A10.

**Questions:**

Can you provide some clarification on Appendix A7:

Are the figures of 2.1% and 0.2% referring to the low-quality rate with the same standard?

Do these numbers represent the rates before and after LLM post-screening?

If so, why is the 2.1% described as "ultimate" when the 0.2% seems to be the final number after post-screening?

---

> ### Author Response · Authors · 2024-11-18
> **Response to Reviewer R2P1**
>
> Thanks for your valuable comments!
> > While the benchmark aims to assess scientific reasoning abilities, it's crucial to acknowledge that QA-like tasks might not be comprehensive enough to evaluate models' scientific reasoning capabilities, even with categorized levels. Assessing models' scientific reasoning capabilities may require more complex setups, including tool usage, task planning, and end-to-end experiment conducting. It's worth mentioning these limitations in order to provide a more nuanced expectation for readers. For example, the authors can add a paragraph or short description as a limitation section, acknowledging that while their benchmark provides valuable insights, fully assessing scientific reasoning may require more complex setups. This would help set more accurate expectations for readers about what the benchmark can and cannot evaluate.
>
> Thank you very much for your thoughtful suggestion! We would like to reiterate that our benchmark aims to evaluate LLMs' capabilities in processing and utilizing **scientific knowledge**, structured across five progressive levels (L1-L5), with L3 focusing on scientific reasoning abilities. While we acknowledge that our evaluation tasks might not be fully comprehensive, we have made significant efforts to cover four scientific domains, 78 tasks, four question types, and 70K evaluation data points.
> That said, we fully agree with your suggestion to acknowledge the need for more complex setups in assessing scientific reasoning. We will include these limitations in the paper to provide a more nuanced expectation for readers and emphasize that our benchmark is a significant step forward but not exhaustive. Additionally, we aim to address these gaps in future work by incorporating more complex setups such as tool usage, task planning, and end-to-end experiment simulations.
>
> > Can you provide some clarification on Appendix A7:
> >
> > Are the figures of 2.1% and 0.2% referring to the low-quality rate with the same standard?
> >
> > Do these numbers represent the rates before and after LLM post-screening?
> >
> > If so, why is the 2.1% described as "ultimate" when the 0.2% seems to be the final number after post-screening?
>
> We apologize for any confusion caused. Specifically, the figures 2.1% and 0.2% indeed refer to the low-quality rates calculated under the same standard, based on the sampled data (5% of the entire dataset).
>
> * The 2.1% figure represents the proportion of low-quality data identified during the first round of human evaluation (i.e., before the LLM post-screening). Here, we use "ultimate" to indicate the end of the first round of "human evaluation" step.
> * The 0.2% figure reflects the low-quality rate obtained after two rounds of human evaluation and LLM post-screening, followed by another round of human evaluation.
>
> We consider 0.2% to be a sufficiently low rate of low-quality data. Given the substantial costs associated with human evaluations, we decided not to proceed with additional evaluation iterations. Therefore, we use "finally" to signify the conclusion of the entire quality control process.

---

> ### Author Response · Authors · 2024-11-25
>
> We sincerely appreciate the time and effort you have dedicated to reviewing our manuscript and providing valuable suggestions. As the author-reviewer discussion phase draws to a close, we would like to confirm whether our responses have effectively addressed your concerns. A few days ago, we provided detailed replies to your comments, and we hope they have adequately resolved the issues you raised. Should you require further clarification or have any additional questions, please do not hesitate to reach out to us. We are more than willing to continue the communication with you. Thank you once again, and we look forward to your further feedback!

---

> > ### Comment · Reviewer_R2P1 · 2024-11-28
> >
> > Thank you for the detailed explanations! The additional lines in the appendix effectively address and eliminate any ambiguity. I also appreciate the authors' efforts to expand the description of the limitations, particularly when addressing 'L5' for 'application,' which could suggest a broader general scope. While this is understandable given the focus remains on evaluation within the domain of knowledge, aligning with the works and datasets cited, the clarification is valuable. In light of these thoughtful revisions and the authors’ dedication to improving the work, I am inclined to raise my rating to reflect these efforts.

---

> > > ### Author Response · Authors · 2024-12-01
> > >
> > > We are delighted to receive your comment! Thank you for your valuable suggestions and feedback!

---

### Official Review · Reviewer_J1QH · 2024-11-03

**Soundness:** 2
**Presentation:** 3
**Contribution:** 2
**Rating:** 5
**Confidence:** 4

**Summary:**

This work introduces a scientific benchmark called SciKnowEval for evaluating Large Language Models (LLMs) in scientific topics. Various sources have been used such as literature, existing QAs, and scientific databases. Based on the generated benchmark, 26 LLMs are evaluated in 5 levels of scientific knowledge which are memory, comprehension, reasoning, discernment, and application.

**Strengths:**

- The paper is well-written and easy to read.
- Extensive examples and experiments are presented.
- The paper provides a valuable benchmark for evaluating LLMs in scientific topics.

**Weaknesses:**

- Various sources, such as literature, existing QAs, and scientific databases, have been used, but the questions included in the dataset are generated from LLMs. This might entail risks of hallucination.
- While methods for evaluating these questions are included, these rely greatly on LLMs, as well. Only 5% of the questions are assessed by a human expert. This might not be sufficient for ensuring safety. Perhaps, it would be helpful that domain experts review a larger sample of the generated questions,
- The experiment results of the 26 LLM comparisons are also rated based on a specific LLM. This can lead to potential biases that could be introduced by using a single LLM for rating.

**Questions:**

- What are some possible limitations of this work?
- How was it determined that 5% was an appropriate sample size for human evaluation?
- Why GPT-4o was chosen to rate LLMs on level 5 - knowledge application? Were there other solutions, such as multiple LLMs, considered?

---

> ### Author Response · Authors · 2024-11-18
> **Responses to Reviewer J1QH (1)**
>
> Thanks for your comments!
> > Various sources, such as literature, existing QAs, and scientific databases, have been used, but the questions included in the dataset are generated from LLMs. This might entail risks of hallucination.
>
> Regarding your concern, we would like to clarify that not all questions in the dataset were generated by LLMs. As mentioned in Section 3.2, Source III ("III. Transforming the Scientific Databases") involves creating new data by using predefined instruction templates and applying them to instructions from scientific databases. In addition, the instruction data for Source I ("I. Generating New QAs from Literature Corpus") is generated based on a segment of related text, with the requirement that the correct answers to these QAs must be accurately derivable from the corpus. Referring to lines 1893-1902 in Appendix A7, we specifically validated the quality of questions and the reliability of answers for Source I, filtering out low-quality data. For Source II ("II. Refactoring the Existing QAs"), the data is derived from high-quality benchmarks. We made only simple adjustments to the question format (e.g., option order) and phrasing, without altering the conditions or choices of the questions. As a result, this process does not introduce significant risks of hallucination.
>
> > While methods for evaluating these questions are included, these rely greatly on LLMs, as well. Only 5% of the questions are assessed by a human expert. This might not be sufficient for ensuring safety. Perhaps, it would be helpful that domain experts review a larger sample of the generated questions,
>
> We understand your concerns. Given that our goal is to construct a large-scale scientific benchmark, it is inevitable that the dataset size will be substantial. Requiring a few domain experts to review a large volume of generated data would result in significant time and financial costs. However, we are committed to maintaining the benchmark regularly.
>
> We will periodically repeat the quality control process as described in Appendix A7. This involves rehiring graduate students and domain experts to sample portions of the data for validation, assess the proportion of low-quality data, and filter out identified problematic data to ensure the overall quality of the benchmark continues to improve.
>
> > The experiment results of the 26 LLM comparisons are also rated based on a specific LLM. This can lead to potential biases that could be introduced by using a single LLM for rating.
>
> We appreciate and understand your concerns about potential biases introduced by using a single LLM for rating. Our approach to using LLM-based evaluation is inspired by well-known LLM leaderboards, such as AlpacaEval [1].
> To clarify, when utilizing GPT-4o for evaluation, we provide it with a reference answer and instruct it to score the evaluated response based on its similarity to the reference. We emphasize that this reference-answer-based assessment enhances the reliability of the evaluation process. We believe that a more powerful model is better equipped to perceive such similarities and provide more reasonable evaluations.
> Additionally, we explored the effectiveness of other LLMs as evaluators, such as Claude-3.5-Sonnet. Our findings show a Pearson correlation of 95.44% between GPT-4o and Claude-3.5-Sonnet, further supporting the consistency and reliability of the evaluation framework.
>
> [1] Dubois Y, Li C X, Taori R, et al. Alpacafarm: A simulation framework for methods that learn from human feedback[J]. Advances in Neural Information Processing Systems, 2024, 36.

---

> ### Author Response · Authors · 2024-11-18
> **Responses to Reviewer J1QH (2)**
>
> Thanks for your comments!
> > What are some possible limitations of this work?
>
> The possible limitations of our work are as follows:
> 1. The use of GPT-4o for evaluation may result in high evaluation costs and potential biases in the assessment process.
> 2. Our evaluation results are not currently easily convertible into scalar values, meaning we can only provide rankings. When new models are added, the rankings need to be recalculated based on the evaluation results of existing models.
>
> > How was it determined that 5% was an appropriate sample size for human evaluation?
>
> First, we must emphasize that our benchmark is of substantial scale, and even a 5% sampling rate involves nearly 4,000 data points. A higher sampling rate would significantly increase the cost of a single round of human evaluation. Second, the primary purpose of human evaluation is to verify data quality and identify types of low-quality data. A 5% sample is sufficient to intuitively reflect the proportion of low-quality data in the benchmark. Furthermore, we have identified specific types of low-quality data through this process and incorporated these findings into prompts, enabling LLMs to automatically filter data with similar issues. Lastly, as mentioned in Appendix A7, we conducted three rounds of sampling (without replacement), bringing the total sampling rate more than 10%.
>
> > Why GPT-4o was chosen to rate LLMs on level 5 - knowledge application? Were there other solutions, such as multiple LLMs, considered?
>
> First, it is important to clarify that all Level 5 questions have standard answers. Using a powerful proprietary model to rate responses **based on these reference answers** is relatively reliable [2]. GPT-4o, being one of the most advanced proprietary models available, meets our requirements effectively. However, we also acknowledge your suggestion of using multiple LLMs to mitigate potential biases. As mentioned on line 510, we are considering training open-source LLM evaluators for assessments (or combining it with GPT-4o and other LLMs for joint evaluation). The advantage of using an open-source model is the significant reduction in evaluation costs.
>
> [2] Kim S, Shin J, Cho Y, et al. Prometheus: Inducing fine-grained evaluation capability in language models. ICLR 2024.

---

> ### Author Response · Authors · 2024-11-25
>
> We sincerely appreciate the time and effort you have dedicated to reviewing our manuscript and providing valuable suggestions. As the author-reviewer discussion phase draws to a close, we would like to confirm whether our responses have effectively addressed your concerns. A few days ago, we provided detailed replies to your comments, and we hope they have adequately resolved the issues you raised. Should you require further clarification or have any additional questions, please do not hesitate to reach out to us. We are more than willing to continue the communication with you. Thank you once again, and we look forward to your further feedback!

---

> ### Author Response · Authors · 2024-12-01
>
> We would like to express our sincere gratitude once again for taking the time and effort to review our manuscript and provide invaluable feedback. As the deadline for the author-reviewer discussion phase has been extended, we apologize for reaching out again to ensure that our responses have sufficiently addressed your concerns. We submitted a detailed reply to your earlier comments a few days ago, and we hope that these responses have effectively resolved the issues you raised. Should you require any further clarification or have additional questions, please feel free to contact us. We would be happy to continue the discussion. Thank you once again, and we look forward to your further feedback.

---

> > ### Comment · Reviewer_J1QH · 2024-12-02
> >
> > Thank you for your response. Having reviewed the authors' responses, I stand by my score.

---

### Official Review · Reviewer_khr7 · 2024-11-03

**Soundness:** 3
**Presentation:** 3
**Contribution:** 3
**Rating:** 6
**Confidence:** 4

**Summary:**

The paper proposes a new SciKnowEval benchmark to evaluate the LLMs across five levels of scientific knowledge, including knowledge memory, knowledge comprehension, knowledge reasoning, knowledge discernment, and knowledge application. The benchmark includes 70k multi-level scientific problems and solutions in the domains of biology, chemistry, and material science. The paper generates benchmarks from three sources: new QAs from the literature corpus, existing QAs, and scientific databases. The paper controls the quality first through LLMs, then through human evaluation, and finally, post-screening by LLMs. The paper tests and ranks 26 LLMs. The paper also shows some interesting findings.

**Strengths:**

1. The paper is clearly written. Each section has figures or tables to help the reader understand how datasets are constructed.
2. The experiment is comprehensive. The paper provides a summary for each level of task. The paper also lists four interesting findings in the discussion section, which can be useful in future research directions.
3. The paper released data and code. In the Appendix, it provides additional results, detailed model descriptions, data sources, and sample prompts.

**Weaknesses:**

1. Some differences between the five levels are not very clear. For example, in Table 3, I did not see a very clear difference between L3 and L4. Additionally, from the definition of section 3.1, it seems that L4 and L5 are very close. The whole level categorization seems to be a little bit artificial. It might be better to create some LLM-based model/human reasoning process instructions to categorize different levels to ensure generalization ability.
2. The paper seems to only provide a ranking for the benchmark. It might be hard to generalize into additional new models.
3. The paper mainly tests 0-shot and 3-shot. Incorporating some additional strategies such as RAG, etc. Given the paper still has half page, the paper can move some of its appendix to the main paper.

**Questions:**

1. How do we determine the level of each task? Are there some standard, replicable instructions?
2. Can you provide more explicit criteria or examples that distinguish between each level?

---

> ### Author Response · Authors · 2024-11-18
> **Response to Reviewer khr7 (1)**
>
> Thank you for your suggestions!
>
> > Some differences between the five levels are not very clear. For example, in Table 3, I did not see a very clear difference between L3 and L4. Additionally, from the definition of section 3.1, it seems that L4 and L5 are very close. The whole level categorization seems to be a little bit artificial. It might be better to create some LLM-based model/human reasoning process instructions to categorize different levels to ensure generalization ability.
>
> We would like to further clarify the differences between L3 and L4. In simple terms, L3 evaluates the reasoning capabilities of LLMs, as reflected in Table 3 by tasks such as numerical computation and function prediction, which are reasoning-focused. In contrast, L4 focuses on the ability to make safe and ethical judgments based on knowledge, such as making decisions that align with safety and ethics or assessing the harmfulness and toxicity of substances. Therefore, L4 emphasizes knowledge discernment, which is significantly different from L3's evaluation of reasoning capabilities.
>
>
> > The paper mainly tests 0-shot and 3-shot. Incorporating some additional strategies such as RAG, etc. Given the paper still has half page, the paper can move some of its appendix to the main paper.
>
> We sincerely appreciate your suggestions! Regarding the RAG strategy you proposed, we will carefully consider its implementation and update the experimental results in future work. As for the suggestions on utilizing the remaining space, we have incorporated improvements in the revised manuscript.

---

> ### Author Response · Authors · 2024-11-18
> **Response to Reviewer khr7 (2)**
>
> Thanks for your comments!
> > How do we determine the level of each task? Are there some standard, replicable instructions?
> > Can you provide more explicit criteria or examples that distinguish between each level?
>
> To provide you with a clearer understanding and judgment of our level settings and decisions, we have further designed the following standards based on the content of the paper:
>
> * **L1**:The core objective of this level is to evaluate the LLMs' ability to memorize scientific knowledge, assessing the breadth and storage of their knowledge.
>     * The differentiation boundary conditions inlcude:
>         * The answers can be explicitly found in the text or database.
>         * The task content is purely memory-based, involving no understanding of context or logical reasoning.
>         * The questions do not involve knowledge points related to safety.
>     * The typical examples include: Any scientific terms, theorems, formulas, timelines, or other explicitly memorizable knowledge points.
>
> * **L2**: The core objective of this level is to assess the LLMs' ability to understand scientific texts or contexts and extract information.
>     * The differentiation boundary conditions inlcude:
>         * The task must include the required contextual text.
>         * The task focuses on understanding causal relationships between paragraphs, inter-sentence relational inference, extraction, and summarization of key content, without involving complex mathematical operations or logical deductions, and unrelated to ethics or safety.
>     * The typical examples include: Scientific text comprehension, experimental data description, simple literature comparisons.
>
> * **L3**: This level evaluates the LLMs' logical reasoning abilities, particularly their capacity to solve scientific reasoning problems.
>     * The differentiation boundary conditions inlcude:
>         * The answers require multi-step reasoning and calculation to obtain.
>         * The questions do not involve safety or ethical judgments.
>         * There must be a standard answer to the problem; any deviation indicates an error.
>         * Questions may include context, such as "Based on experimental data, determine whether a chemical reaction complies with the second law of thermodynamics", but they must require multi-step reasoning and calculation.
>     * The typical examples include: Quantum mechanics formula derivation, biological evolution process calculation, chemical reaction pathway prediction, hypothesis testing.
>
> * **L4**: This level assesses the LLMs' ability to discern scientific ethics, safety, and decision-making.
>     * The differentiation boundary conditions inlcude:
>         * The tasks involve harmful substances or harmful operations.
>         * The tasks involve judgments of safety risks.
>     * The typical examples include: Experimental safety evaluation, toxic substance prediction, misuse scenario analysis.
>
> * **L5**: The core objective of this level is to test the LLMs' ability to apply scientific knowledge in innovative ways to real-world scenarios.
>     * The differentiation boundary conditions inlcude:
>         * The task must involve a problem or design that is operable in the real world.
>         * The questions may not have absolute standard answers; evaluations are based on relative reliability (e.g., similarity to reference answers) rather than direct accuracy calculations.
>     * The typical examples include: Biological experiment design, molecular synthesis planning, equipment optimization schemes.

---

> > ### Comment · Reviewer_khr7 · 2024-11-21
> >
> > Thank you very much for your reply! Based on your response and other reviewers' comments, I still do not think the difference between L3 and L4 is clear enough, and L4 should be included as part of the conditions. Therefore, I decided to keep my score.

---

> > > ### Author Response · Authors · 2024-11-25
> > >
> > > We sincerely thank you for your valuable insights and thoughtful feedback!

---

### Official Review · Reviewer_epnp · 2024-11-04

**Soundness:** 3
**Presentation:** 3
**Contribution:** 2
**Rating:** 5
**Confidence:** 3

**Summary:**

This paper constructs a benchmark to evaluate LLM's scientific knowledge and tests many LLM's performance on the benchmark.
Specifically, the benchmark is designed based on five levels: Studying extensively, Enquiring earnestly, Thinking profoundly, Discerning clearly, and Practicing assiduously. The five levels are based on an ancient Chinese book.

The benchmark is collected from existing literatures, existing QAs, and science databases.

**Strengths:**

1. This is a multi-level benchmark covering multiple disciplines. Thousands to tens of thousands of questions are collected at each level. It could be comprehensive.
2. This paper compares many LLM's performance on the benchmark.

**Weaknesses:**

1. Although with a good starting point, it is unclear whether the five levels listed in this paper truly reflect how humans think and understand this world. The five levels are from an old Chineses book, but it is unclear whether it has been tested by modern cognitive science. I would suggest the authors to find cognitive science findings to support to classification into the five levels.
2. It is not persuading how the data collected can fully represent the five levels. Particularly, level 4 is only about safety. But is safety a necessary step for the learning process? It seems the 5 levels are a mixture of different requirements, but not focused on reasoning levels. I would suggest the authors to provide a clear difference on the five levels, and also the why it is necessary to classify them to five levels: there's no need to add more complexity if it is not necessary.
3. The insights are a bit limited from the paper. The discussions in section 4.3 do not seem surprising. It is unclear what knowledge can be learned from the paper. For example, a takeaway knowledge could be how fundamentally are the difficulties and challenges differ in five levels? And LLMs can be adapted to deal with the challenges?

**Questions:**

See above

---

> ### Author Response · Authors · 2024-11-19
> **Response to Reviewer epnp (1)**
>
> Thanks for your comments!
> > Although with a good starting point, it is unclear whether the five levels listed in this paper truly reflect how humans think and understand this world. The five levels are from an old Chineses book, but it is unclear whether it has been tested by modern cognitive science. I would suggest the authors to find cognitive science findings to support to classification into the five levels.
>
> To address your concerns, we revisited modern cognitive science frameworks for categorizing cognition. Among these, Bloom's taxonomy [1], a renowned framework for classifying educational objectives, divides cognitive learning goals into six levels:
>
> 1.**Remember**: Recognizing or recalling facts, terms, basic concepts, or answers without necessarily understanding their meaning. We believe this aligns with the goal of Studying Extensively (L1) in our work.
>
> 2.**Understand**: Demonstrating an understanding of facts and ideas by organizing and summarizing information. This matches the tasks in Enquiring Earnestly (L2), such as Text Summary, Hypothesis Verification, and Relation Extraction. These tasks require answering based on a text rich in information through comprehension, organization, and summarization.
>
> 3.**Apply**: Using acquired knowledge to solve problems in new or unfamiliar situations. This corresponds to the intent behind Practicing Assiduously (L5), though our L5 focuses more on real scientific research scenarios, aiming at solving complex scientific problems and creating innovative solutions.
>
> 4.**Analyze**: Breaking down information into parts to understand relationships, motives, or causes. This resonates with Thinking Profoundly (L3), which goes further by assessing logical deduction, numerical computation, and functional prediction capabilities.
>
> 5.**Evaluate**: Making judgments about information based on set criteria or standards. In our framework, Discerning Clearly (L4) emphasizes evaluating LLMs' ability to discern harmful and toxic information, which falls under the domain of "making judgments".
>
> 6.**Create**: Building a new whole by combining elements or creating new meaning. This is also closely aligned with our L5.
>
> Although the interpretations and classification methods are not identical, we find that the wisdom and philosophy of ancient thinkers align remarkably well with modern cognitive science, demonstrating their rationality and relevance. Finally, it is important to clarify that the five levels we designed are **not intended to fully replicate the human cognitive process but rather to provide an effective method for task stratification**.
>
> [1] https://en.wikipedia.org/wiki/Bloom%27s_taxonomy

---

> ### Author Response · Authors · 2024-11-19
> **Response to Reviewer epnp (2)**
>
> Thanks for your comments!
> > It is not persuading how the data collected can fully represent the five levels. Particularly, level 4 is only about safety. But is safety a necessary step for the learning process? It seems the 5 levels are a mixture of different requirements, but not focused on reasoning levels. I would suggest the authors to provide a clear difference on the five levels, and also the why it is necessary to classify them to five levels: there's no need to add more complexity if it is not necessary.
>
> We emphasize that it is difficult–if not impossible–to design tasks that can fully represent the five levels. Instead, we propose this constructive evaluation framework as a first step and strive to collect and design as many tasks as possible to reflect different capabilities. Nevertheless, we have collected 78 tasks across four scientific domains, comprising a total of 70K data points. To the best of our knowledge, this constitutes the largest and most comprehensive scientific capability evaluation benchmark to date.
>
> We reiterate that **Level 4 (L4) evaluates the LLM’s ability to make correct, secure, and ethical decisions based on scientific knowledge**. This includes assessing the harmfulness and toxicity of information and understanding the ethical implications and safety concerns related to scientific endeavors. This type of evaluation reflects the model’s judgment and critical thinking capabilities, which are indispensable during the learning process.
>
> Furthermore, we would like to clarify that the five levels represent five distinct capabilities of LLMs in processing and utilizing knowledge, rather than levels of reasoning. In simpler terms:
> * L1 corresponds to basic knowledge memory,
> * L2 to contextual understanding,
> * L3 to analytical reasoning,
> * L4 to knowledge discernment, and
> * L5 to knowledge creation and application.
>
> The advantage of designing these different levels lies in their ability to highlight the strengths and weaknesses of models across varying levels, facilitating targeted improvements. For instance:
>
> * weaknesses at L1 might reflect deficiencies in pretraining knowledge, which can be mitigated through Retrieval-Augmented Generation (RAG) or continued pretraining.
> * Shortcomings at L3 may indicate gaps in reasoning abilities, which can be enhanced using Chain-of-Thought (CoT) prompting.
> * Deficiencies at L4 might suggest a lack of safety awareness, which can be addressed through techniques like Direct Preference Optimization (DPO) or Reinforcement Learning with Human Feedback (RLHF).
>
> We believe that such fine-grained categorization is necessary to thoroughly assess each model's strengths and identify areas for further improvement.
>
> To enhance readers' comprehension of our framework, we will incorporate the aforementioned information into the revised paper, providing a clear difference on the five levels.
>
>
> > The insights are a bit limited from the paper. The discussions in section 4.3 do not seem surprising. It is unclear what knowledge can be learned from the paper. For example, a takeaway knowledge could be how fundamentally are the difficulties and challenges differ in five levels? And LLMs can be adapted to deal with the challenges?
>
> Thank you for your valuable suggestions! The discussions in Section 4.3 aim to demonstrate that our benchmark can validate these intuitive phenomena, such as showing that our benchmark has sufficient discriminative power to reflect performance differences between small and large models, as well as between newer models (e.g., o1) and older ones. Additionally, we have demonstrated several methods to improve performance in scientific domains, such as few-shot learning and instruction fine-tuning. Furthermore, as a benchmark-oriented paper, we have highlighted in the analysis of results in Section 4.2 several areas where LLMs fall short at each level.
>
> Lastly, your suggestion about exploring “how fundamentally are the difficulties and challenges differ in five levels” is a constructive point. While we have already speculated on potential reasons for model failures in the results analysis of Section 4.2, we believe that the fundamental differences lie in the distinct capabilities assessed at each level. For example, o1 significantly enhanced reasoning ability through implicit chain-of-thought (CoT) and improved safety through integrating safety rules into its reasoning process. These improvements are evident in L3 and L4, as shown in Figure A1 (Page 44). However, we did not observe significant performance gains in the other three levels.
>
> We plan to conduct further experiments to more explicitly analyze the fundamental challenges of tasks at the five levels and validate how models can be optimized to overcome existing capability bottlenecks.

---

> ### Author Response · Authors · 2024-11-25
>
> We sincerely appreciate the time and effort you have dedicated to reviewing our manuscript and providing valuable suggestions. As the author-reviewer discussion phase draws to a close, we would like to confirm whether our responses have effectively addressed your concerns. A few days ago, we provided detailed replies to your comments, and we hope they have adequately resolved the issues you raised. Should you require further clarification or have any additional questions, please do not hesitate to reach out to us. We are more than willing to continue the communication with you. Thank you once again, and we look forward to your further feedback!

---

> ### Author Response · Authors · 2024-12-01
>
> We would like to express our sincere gratitude once again for taking the time and effort to review our manuscript and provide invaluable feedback. As the deadline for the author-reviewer discussion phase has been extended, we apologize for reaching out once more to confirm whether our responses have adequately addressed your concerns. We submitted a detailed reply to your earlier comments several days ago, and we hope that these responses have sufficiently resolved the issues you raised. If you require any further clarification or have additional questions, please do not hesitate to contact us. We are more than happy to continue the conversation. Thank you once again, and we look forward to your further feedback!

---

> ### Comment · Reviewer_epnp · 2024-12-02
>
> Thank you for the response.
>
> "the five levels we designed are not intended to fully replicate the human cognitive process but rather to provide an effective method for task stratification.":
>
> The same question arise again: why does the paper stratify the tasks into the five levels? I think it should stratify based on science findings. Cognitive science is just a possible source. Otherwise what benefits can the community have with the stratification? If there is some benefit, then probably there will be some (cognitive) science backup.
>
> I would also suggest to not find a wikipedia page as reference, but at least a published paper.
>
> If the stratification is meaningful, under the circumstance that only a partial of each layer can be represented with the benchmark, I would suggest the authors to analyze what are the challenges of the benchmark, as it is designed, but not just how LLMs perform on it. Currently the field are not in short of benchmark, given the existing of LLMs that can help with the construction, but meaningful benchmark that can let the users know, what does it mean to reach a certain performance at the benchmark, and what insights can the users obtain after they have tested their models in the benchmark.

---

> > ### Author Response · Authors · 2024-12-02
> >
> > Thank you for your valuable feedback and suggestions!
> >
> > > Why does the paper stratify the tasks into the five levels? I think it should stratify based on science findings. Cognitive science is just a possible source.
> >
> > Firstly, we emphasize that the purpose of task stratification is to comprehensively examine the LLM’s abilities at different cognitive levels, **just as a teacher designs exams that include questions at different levels to assess students' performance across various cognitive skills (human preference alignment)**. We believe that by stratifying the tasks and focusing on different cognitive abilities, we can better reveal the strengths and weaknesses of LLMs at each level, thereby promoting continuous improvement and development of the models.
> >
> > In our design, we adopted the philosophical framework of the “Doctrine of the Mean” from ancient Chinese philosophy, which has guided Chinese education for thousands of years and shares many similarities with the application of Bloom's Taxonomy in education. **Bloom's Taxonomy is widely used in exam design (similar to benchmark design) and educational contexts to effectively assess students' performance at different cognitive levels [1][2]**. We believe this framework helps to comprehensively examine LLMs’ cognitive abilities and provides the community with an easy-to-understand stratification framework. Specifically, compared to Bloom's Taxonomy, we use a more simplified five-level design that concisely reflects LLMs' performance at each cognitive level and provides a clear direction for improvement.
> >
> > > If the stratification is meaningful, under the circumstance that only a partial of each layer can be represented with the benchmark, I would suggest the authors to analyze what are the challenges of the benchmark, as it is designed, but not just how LLMs perform on it.
> >
> > We greatly appreciate the reviewer’s valuable suggestion. Although our designed benchmark does not cover all tasks for each cognitive level, we believe that by selecting representative tasks, we can effectively reflect LLMs' abilities across different levels. If the number of tasks were sufficient, it could better fit the performance of each level, thus helping to evaluate the model’s cognitive capabilities.
> >
> > Regarding the challenges of the benchmark design, we emphasize the following points:
> >
> > 1. **Challenge of Lack of Task Classification**: We found that existing well-known benchmarks (such as MMLU, SciQ, SciBench, etc.) cover a wide range of tasks but do not categorize them, making it difficult to accurately assess where LLMs fall short. To help the community better pinpoint these shortcomings for targeted improvements, we adopted a task stratification framework inspired by the "Doctrine of the Mean". This framework, based on thousands of years of educational experience, helps us clearly differentiate tasks at different levels and guide researchers in identifying areas where LLMs need improvement.
> >
> > 2. **Challenge of Task and Data Quantity**: Although recent benchmarks (such as SciAssess, ChemBench) have implemented task stratification, due to the lack of clear classification standards and sufficient task numbers, these benchmarks do not effectively showcase LLMs’ abilities at each cognitive level, leading to limited fitting for the levels. Therefore, we supplemented our task classification standards in Appendix A12. In terms of task design, we aimed for greater precision and ensured task quantity and data coverage with three data construction methods (as described in Section 3.2). We ultimately built 78 scientific tasks and 70K evaluation data, the largest and most comprehensive dataset in the field.
> >
> > 3. **Significance of Multi-Level Evaluation**: We believe that SciKnowEval’s multi-level evaluation is one of its most important features. Through this evaluation, researchers can precisely identify LLMs’ performance across cognitive levels (L1-L5). For example, models that perform poorly at the L1 level may indicate issues with knowledge injection during pre-training, while weaknesses at the L3 level reflect limitations in reasoning abilities. By analyzing poorly performing tasks in specific level, researchers can identify domains and scenarios that need improvement. This detailed analysis of specific tasks allows researchers to clearly recognize the strengths and limitations of the models and further improve their performance on specific cognitive tasks.
> >
> > We believe this detailed multi-level evaluation can provide valuable insights for researchers, helping them to more effectively optimize and enhance LLMs’ cognitive capabilities.
> >
> > ---
> >
> > [1] Omar N, Haris S S, Hassan R, et al. Automated analysis of exam questions according to Bloom's taxonomy. Procedia-Social and Behavioral Sciences, 2012, 59: 297-303.
> > [2] Chandio M T, Pandhiani S M, Iqbal R. Bloom's Taxonomy: Improving Assessment and Teaching-Learning Process. Journal of Education and Educational Development, 2016, 3(2): 203-221.

---

### Author Response · Authors · 2024-11-23
**General Response to Reviewers and Revision Submitted**

We sincerely thank all reviewers for their insightful comments and suggestions! We have revised the paper to address the reviewers' concerns. Below, we summarize the major revisions, while we have also provided responses to each reviewer's comments separately.

The main revisions are as follows:
1. In Appendix A7, we added additional descriptions related to quality control to address potential questions from readers. (Reviewers J1QH, R2P1)
2. In Appendix A9, we provided more details about using GPT-4o for evaluation. Specifically, we emphasized that incorporating reference answers during the evaluation process makes the results more reliable. (Reviewer J1QH)
3. In Appendix A12, we included detailed task categorization criteria to enable accurate classification across different levels. (Reviewers epnp, khr7, J1QH)

---

### Comment · Area_Chair_YonA · 2024-11-27

Dear reviewers,

Thank you for your efforts reviewing this paper. If you haven't, can you please check the authors' responses and see if your concerns have been addressed? Please acknowledge you have read their responses. Thank you!

---

### Meta-Review · Area_Chair_YonA · 2024-12-22

**Metareview:**

Summary:

This paper introduces the SciKnowEval benchmark to evaluate LLMs across five progressive levels of scientific knowledge (studying extensively, inquiring earnestly, thinking profoundly, discerning clearly, and practicing assiduously), which aim to assess the breadth and depth of scientific knowledge in LLMs (e.g., memory, comprehension, reasoning, discernment, and application). The paper constructs a large-scale evaluation dataset encompassing 70K multi-level scientific problems and solutions in the domains of biology, chemistry, physics, and materials science. This paper benchmarks 26 open-source and proprietary LLMs using zero-shot and few-shot prompting strategies. The results reveal that despite the state-of-the-art performance of proprietary LLMs, there is still significant room for improvement, particularly in addressing scientific reasoning and applications.

Strengths:

1. The proposed benchmark covers multi-level scientific knowledge in multiple disciplines.

2. The experiments are comprehensive in the sense that the paper compares many LLMs’ performance.

Weaknesses:

The following are outstanding issues after the discussion period:

1. The paper lacks connections with modern cognitive science or support from other scientific literature that the five levels truly reflect how humans think and understand this world. The five levels seem to be a mixture of different things and it is not clear why it’s necessary to have five levels or the differences between certain levels are not clear. (Reviewer epnp and khr7)

2. Deep insights about the challenges of the benchmark other than the low performance of LLMs are lacking. (Reviewer epnp)

3. Only a small portion of questions (5%) are assessed by a human expert. (Reviewer J1QH)

**Additional Comments On Reviewer Discussion:**

Issues that have been addressed (at least to a large extent) are summarized as follows:

1. Reviewer J1QH raised many concerns including the potential bias of using one specific LLM, questions in the dataset being generated by LLMs, limitations of the work, etc. Most of these concerns were addressed after the discussion period.

2. Reviewer R2P1 raised a few clarification questions such as the limitations of this work (assessing models’ scientific reasoning capabilities would require more complex setups), which were addressed by the author responses.

---

### Decision · Program_Chairs · 2025-01-22

Reject